# Field Trials of Live and Inactivated Camelpox Vaccines in Kazakhstan

**DOI:** 10.3390/vaccines12060685

**Published:** 2024-06-19

**Authors:** Muratbay Mambetaliyev, Sanat Kilibayev, Marzhan Kenzhebaeva, Nuraiym Sarsenkulova, Shalkar Tabys, Aisulu Valiyeva, Dias Muzarap, Moldir Tuyskanova, Balzhan Myrzakhmetova, Nurkuisa Rametov, Aizhamal Sarbassova, Ryspek Nurgaziev, Aslan Kerimbayev, Shawn Babiuk, Kuandyk Zhugunissov

**Affiliations:** 1Research Institute for Biological Safety Problems, Gvardeiysky 080409, Kazakhstan; m.mambetalyev@biosafety.kz (M.M.); sanat.kilibaev@mail.ru (S.K.); m.kenjebaeva@biosafety.kz (M.K.); n.sarsenkulova@biosafety.kz (N.S.); sh.tabys@biosafety.kz (S.T.); a.dulatbekovna@biosafety.kz (A.V.); d.muzarap@biosafety.kz (D.M.); m.serzhankyzy@biosafety.kz (M.T.); b.myrzakhmetova@biosafety.kz (B.M.); a.kerimbayev@biosafety.kz (A.K.); 2Institute of Ionosphere, Almaty 050020, Kazakhstan; nurkuisa.rametov@gmail.com; 3Department of Geospatial Engineering, Satpayev Kazakh National Research Technical University, Almaty 050013, Kazakhstan; 4Department of Social Sciences, Satpayev Kazakh National Research Technical University, Almaty 050013, Kazakhstan; aizhamalsarbassova@gmail.com; 5Department of Infectious Animal Diseases, Skryabin Kyrgyz State Agrarian University, Bishkek 720000, Kyrgyzstan; rysbekn@mail.ru; 6National Centre for Foreign Animal Disease, Canadian Food Inspection Agency, Winnipeg, MB R3E 3M4, Canada; shawn.babiuk@inspection.gc.ca; 7Department of Immunology, University of Manitoba, Winnipeg, MB R3T 2N2, Canada

**Keywords:** camelpox, field trial, live vaccine, inactivated vaccine, scarification, safety, immunogenicity

## Abstract

An outbreak of camelpox occurred in the Mangistau region of Kazakhstan in 2019. To control the outbreak of camelpox and to prevent its further spread to other regions, camels were vaccinated using live and inactivated camelpox vaccines produced in Kazakhstan. To evaluate the efficacy of these camelpox vaccines in the field, vaccine trials used 172 camels on camel farms in the Beineu district. Of these, 132 camels were vaccinated using a live attenuated camelpox vaccine and 40 camels were vaccinated using an inactivated vaccine to observe immunogenicity and safety. The live vaccine was inoculated into camels by scarification at a dose of 5 × 10^4^ EID_50_, and the inactivated vaccine was injected intramuscularly at 5 mL twice, with an interval of 35 days. During the safety evaluation, camels administered either vaccine displayed no clinical signs of illness or any adverse effects. Post-vaccination seroconversion demonstrated that the live attenuated vaccine started to elicit antibody responses in some animals as early as day seven, while, by day 28, 99% of vaccinated camels responded. For camels immunized with the inactivated vaccine, seroconversion began on day 21 at low titers ranging from 1:2 to 1:4. Ninety days post vaccination, 77% of the camels demonstrated an immune response that was up to a titer of 1:16. The antibody response waned six months post vaccination in camels vaccinated with two types of vaccine. Nonetheless, both vaccines were 100% effective at preventing clinical disease in vaccinated camels during the camelpox outbreak. All unvaccinated camels became ill, with manifestations of clinical signs characteristic of camelpox. Following these successful field trials in Kazakhstan, a vaccination program for camels, to control camelpox using the domestically produced live attenuated camelpox vaccine, has started.

## 1. Introduction

Camel farming is a vital part of livestock agriculture within the desert and semi-desert regions of Kazakhstan. It fulfills the local population’s demand for meat, wool, and milk while promoting development in these areas [1]. According to the Statistics Committee of Kazakhstan’s Ministry of National Economy, in 2021, the camel population exceeded 243.4 thousand [2], primarily in the southern and western regions of the country.

Various infectious diseases including camelpox wreak havoc on the industry and hinder camel production [3]. Camelpox is a contagious viral ailment that causes fever, skin rashes, abortions, and even fatalities in camels. Camelpox virus (CMLV), which belongs to the Orthopoxvirus (OPV) genus of the Poxviridae family, causes the disease [4]. Young animals are more susceptible to camelpox compared to older camels, with a mortality rate reaching up to 30% [4,5].

A diagnosis of camelpox must be distinguished from various diseases, such as necrobacillosis, foot and mouth disease, fungal skin infections, scabies, contagious ecthyma, papillomavirus infection, brucellosis and arthropod bites. Virus isolation and PCR-based diagnostics can detect the camelpox virus or genome, which is required for laboratory confirmation of the disease. The virus neutralization test (VNT) and ELISA diagnostics can detect camelpox-virus-specific antibodies. All orthopoxviruses exhibit serological cross-reactivity within their genus [6,7]. The parapoxviruses and papillomaviruses that infect camels do not show cross-reactivity with the camelpox virus, facilitating their differentiation from camelpox [8]. There is no specific treatment for camelpox, although some antiviral drugs have proven to be moderately effective in animal models [9]. Both live attenuated and inactivated vaccines are effective at controlling camelpox [10]. However, this requires vaccines to be available and the camels to be vaccinated. It can be difficult to procure camelpox vaccines in various countries for numerous reasons, including regulatory issues. Having locally produced, available camelpox vaccines accepted for use by the countries’ regulatory agencies provides them with the capability to vaccinate camels.

Camelpox occurs in almost every camel-farming country [11]. In 2017–2021, camelpox outbreaks occurred in 12 countries worldwide [12]. When analyzing camelpox cases over the past eight years, it was found that four countries (Israel, Iraq, Eritrea, Kazakhstan) had sporadic cases, while the remaining eight countries (Iran, Libya, Oman, Palestine, Saudi Arabia, Somalia, Tunisia, Ethiopia) were endemic for this disease [12,13].

Significant camelpox outbreaks have occurred in Kazakhstan since 1930. Notably, outbreaks occurred in the Mangystau and Atyrau regions of the Kazakh Soviet Socialist Republic in 1930, 1942–43, and 1965–1969 [14]. After the collapse of the USSR, the last major outbreak of camelpox occurred in three districts (Mangistau, Tupqaragan, and Qaraqiya) of the Mangistau region in March 1996 [15]. This outbreak in 8000 camels caused 43 mortalities and 830 camels to become clinically ill [16].

During this outbreak of camelpox in 1996, an epizootic strain M-96 was isolated from the Mangistau region. The complete genome sequencing of the virus (accession number AF438165.1), confirmed its origin as being from the Iranian strain [17]. Epidemiological data on the incidence of camelpox in the Republic of Kazakhstan made it possible to establish the cyclical occurrence of epizootics, which is approximately 23 years. Repeated outbreaks of camelpox in Kazakhstan, caused by the presence of camelpox virus in the region, demonstrate the need for a domestically available camelpox vaccine, which was not available. Due to the lack of a domestic vaccine for the prevention and control of the epizootic strain M-96 of the CMLV, the attenuated strain KM-40, derived from the camelpox wild-type strain M-96 through 40 serial passages on the chorioallantoic membrane of chicken embryos, was developed. A live attenuated and inactivated domestic camelpox vaccine using the attenuated strain KM-40 were developed. It has been previously demonstrated that this live attenuated camelpox vaccine is safe by overdose inoculation, and it does not revert to virulence following three successive passages in camels [18]. In addition, the protective effectiveness of both the live attenuated, including the optimal dose to use, administered through scarification, and inactivated camelpox vaccines were evaluated using an experimental infection of camelpox in camels to test their efficacy [18,19].

In June 2019, 23 years after the previous outbreak, a camelpox epidemic began in the Qaraqiya district of the Mangystau region (Figure 1) [20]. The epidemic initially started with isolated cases and gradually spread to all closely associated animals irrespective of age, sex, or physiological state. The epidemic later extended to multiple districts in the Mangystau region, affecting 2336 camels, with 27 confirmed fatalities. Additionally, there were reports of 390 stillborn and miscarried camels.

These vaccines were able to control the camelpox outbreak in the Mangystau region and establish a buffer zone in border areas. This article reports these vaccination results, obtained under field conditions in Kazakhstan.

## 2. Materials and Methods

### 2.1. Preparation of Live Vaccines

The KM-40 strain of the camelpox virus was cultivated in the chorioallantoic membrane of chicken egg embryos, as previously described [18]. The formulation of the live attenuated vaccine included adding a stabilizer (13% peptone) to the virus suspension and subsequently lyophilizing it.

### 2.2. Preparation of the Inactivated Vaccine

The CMLV KM-40 strain was cultivated on lamb kidney cells [19] using Dulbecco’s modified Eagle’s medium (DMEM) supplemented with 10% irradiated fetal calf serum. Lamb kidney cells were infected with the virus at a multiplicity of infection (MOI) of 0.01 TCID_50_/cell. The infected cells were incubated at 37 °C for 1 h. After adsorption for 1 h, DMEM with 2% fetal calf serum was added to the infected cells. Virus-infected cells were incubated at 37 °C for 7 days [21]. When 80–90% of the cell monolayers exhibited 80–90% CPE, they were frozen at −40 °C for 16–18 h. Next, the viral material was thawed at 22 ± 1 °C and combined into one sterile vessel. The viral titer was determined according to the Reed and Muench method [22].

The camelpox virus titer 6.25 ± 0.08 lg TCID50/mL was inactivated using 98% β-propiolactone at a final concentration of 0.05% to produce the inactivated vaccine antigen, similar to inactivated capripoxvirus vaccines [23]. Virus isolation verified that the vaccine virus was inactivated and that the product was determined to be sterile prior to formulation. The inactivated vaccine antigen was adjuvanted with aluminum hydroxide at a concentration of 1.25 mg/mL. The finished vaccine was stored in 50 mL vials at 4 °C until needed [19].

### 2.3. Study Site and Baseline Serological Survey

The Mangystau region, which is situated in the southwestern part of Kazakhstan, shares borders with Azerbaijan and Iran in the west, across the Caspian Sea, Turkmenistan in the south, Uzbekistan in the southeast, and the Atyrau and Aktobe regions of Kazakhstan in the north and northeast. This region’s landscape comprises salt marshes and sandy areas, which are mostly devoid of vegetation. The climate here is primarily continental, with dry, scorching summers and brief winters. The annual precipitation levels range between 100 and 150 mm.

Field trials of the candidate vaccines occurred between 3 February and 15 August 2020. The vaccination trials used candidate vaccines in three rural areas of the Beineu district in the Mangystau region, namely, Beineu, Borankul, and Sam (Figure 1). There are 9230 camels in Beineu, 2854 in Borankul, and 2813 in Sam. An epidemiological assessment of camelpox occurred in these districts prior to starting vaccine testing. This comprehensive survey involved interviewing veterinarians and animal owners, conducting clinical examinations, and collecting camel serum samples. The prevalence of camelpox was assessed among the camels in the study area. Randomly selected herds and camels—totaling up to 260 camels from five herds—were chosen for the study. These herds resided within a 10–100 km radius of the site of a camelpox outbreak. The serum samples were heat-treated at 56 °C for 30 min to inactivate any other pathogens. The treated serum samples were frozen and sent to the Research Institute for Biological Safety Problems for the analysis of camelpox virus using a virus neutralization test (VNT).

### 2.4. Protocol for the Vaccination of Camels and Study Design

A total of 117 camels of varying ages were vaccinated with the live attenuated vaccine. Prior to vaccination, the hair on the left side of each camel’s neck was shaved and cleaned using 96% ethanol. The lyophilized vaccine was reconstituted with 50% glycerol in phosphate-buffered saline and was administered at a final volume of 0.2 mL and a field dose of 5 × 10^4^ EID_50_. The vaccine was administered using scarification on the previously prepared neck skin (Table 1).

The inactivated vaccine was administered as an intramuscular injection of 5 mLs to 30 young camels aged 5–9 months old. A second vaccination was administered 35 days later. It is important to mention that other camels on the same farms were unvaccinated to serve as controls (Table 1). The monitoring of camels following their live or inactivated vaccination occurred daily for 14 days to watch for potential symptoms of camelpox, such as elevated body temperature, characteristic clinical signs, and inflammation at the injection site. Following vaccination, serum samples were collected weekly for the first month and at 90 and 180 days. All camels, both vaccinated and unvaccinated, grazed together within a 15–20 km radius of the farm.

### 2.5. Safety

To evaluate the safety of the live attenuated vaccine, an overdose of the vaccine was performed on 15 camels of varying ages, including five camels in their last trimester of pregnancy (Table 1). The vaccine was administered in phosphate-buffered saline (PBS) in a 1 mL volume at a dose of 10^6^ EID50/mL, injected subcutaneously to the neck region. The inactivated vaccine was administered intramuscularly to ten younger camels in a 10 mL volume. Vaccinated camels were monitored daily for their general health, temperature, and any deviations from physiological norms.

### 2.6. Virus Neutralization Test (VNT)

The VNT was performed according to the WOAH terrestrial manual (2012) [24]. This process involves the use of a strain of CMLV and specific sera for this strain. The sera were diluted from 1:2 to 1:128 and then combined in equal volumes with the CMLV strain at a concentration of 100 TCID_50_/_mL_.

### 2.7. Delayed-Type Hypersensitivity Test (DTH)

The DTH procedure was conducted based on the technique outlined in reference [25]. This process utilized an antigen developed from the “KM-40” strain of CMLV cultivated in lamb kidney cells.

All vaccinated and control camels received an intradermal injection, in the shaved area on the left side of their neck, of 0.2 mL of heat-inactivated KM-40 antigen from the CMLV strain. The skin thickness at the injection site was measured every two days over a five-day period using a caliper. This served as an indicator of hypersensitive reactions.

### 2.8. Statistical Analyses

We utilized the GraphPad Prism 9 program (GraphPad Software, Inc., San Diego, CA, USA) for the statistical analysis of the research findings. This study involved the application of descriptive statistics to examine all the data and calculate mean values (M) and standard deviations (SD). Student’s *t*-test was used to evaluate the statistical significance of the differences between the vaccinated and unvaccinated groups in terms of their DTH. Statistically significant values were those with *p* ≤ 0.05. 

The spatial processing functionality of ArcGIS pro 3.3 (ESRI, Redlands, California, USA) was used to perform overlay analysis to create buffer zones around populated vaccine trial sites. Spatial outbreak data was stored in a geodatabase to create vector maps.

### 2.9. Ethics Statement

The animal-related work was conducted in accordance with the Laws for Responsible Animal Handling (Law No. 97-VII ZRK, Republic of Kazakhstan, 30 December 2021) and other applicable guidelines. The Biological Safety Problems Research Institute’s Bioethics Review Board approved the study plans, with permit number 0501/020, before the study began. Institutional codes, operating procedures, and animal care guidelines were upheld throughout the research.

## 3. Results

### 3.1. Results of Epidemiological Surveys and Local Seroprevalence

According to the Mangystau Regional Territorial Inspectorate, camelpox outbreaks occurred in June 2019 in the Qaraqiya, Mangystau, Munaily, and Tupqaragan districts of the Mangystau region. During this epidemic, 3.7% of the camel population in these four districts contracted the disease. Among those in the infected group, 3.4% were young camels aged 5–7 months, while 96.5% were adults, and, within this group, 1.2% experienced abortion. Moreover, in the three rural districts of the Beineu region where our research is ongoing, no reports of camel diseases emerged despite their proximity to the outbreak. Consequently, these regions are relatively disease-free from an epidemiological perspective. However, three camels in the border area of the Beineu district adjacent to the Mangystau region contracted camelpox (See Appendix A).

In the initial study, none of the serum samples collected from 260 camels in these three rural districts tested positive for camelpox by VNT (Table 2).

### 3.2. Safety of Live and Inactivated Vaccines

The average body temperature recorded for both young and pregnant camels prior to vaccination was 37.9 °C, with an SD of 0.2 and a range from 37.7 to 38.1 °C. The temperatures for pregnant camels were slightly higher, averaging 38.7 °C, with an SD of 0.5 and a range from 38.5 to 38.7 °C. Regardless of age, the camels that received a live vaccine did not display any temperature increase during observation. However, pregnant camels exhibited a 0.4 to 0.8 °C higher body temperature than their nonpregnant counterparts (Figure 2). All camels that received the inactivated vaccine also maintained a consistent body temperature during the observation period, with two exceptions: one camel displayed a minor temperature increase to 39.9 °C on the third day, while the other displayed an increase to 38.8 °C on the fifth day. Nonetheless, for the most part, the body temperature of the camels remained within standard physiological parameters (Figure 2). Furthermore, following vaccination, whether with a live or inactivated vaccine, no side effects or adverse clinical symptoms occurred in any of the camels.

### 3.3. Serological Response

Table 3 shows the antibody responses, determined using VNT, in camels at different time intervals following the administration of the two vaccines. In camels vaccinated with the live attenuated vaccine, 4 out of the 117 camels (approximately 3.42%) developed VNT responses at a titer of 1:2 seven days post vaccination (dpv). At 14 dpv, antibodies with a titer of 1:2 were detected in 13 camels (equivalent to 11.1%), while 2 others (representing 1.71%) had a titer of 1:4. At 21 dpv, 53 camels (45.3%) had a titer of 1:2, 34 (29.1%) had a titer of 1:4, and 21 (17.9%) had a titer of 1:8. On 28 dpv, 13 camels (11.1%) had a titer of 1:2, while 58 (49.5%) and 45 (38.4%) had antibody titers of 1:4 and 1:8, respectively, indicating that 99% of the vaccinated camels seroconverted. At 90 dpv, the antibody levels in vaccinated camels had risen to between 1:4 and 1:128, with 99% of the camels demonstrating seroconversion. On 180 dpv, their antibody levels decreased, with responses peaking at a 1:8 dilution, and there was an 8.4% decrease in the number of camels demonstrating seroconversion.

The antibody responses in camels vaccinated with an inactivated vaccine were also assessed using the VNT. The peak levels of VNT responses never exceeded a ratio of 1:16 at all time points, except for 7 and 14 dpv. On 21 dpv, nearly 27% of the camels showed an immune response, which increased to approximately 33% on 28 dpv. By 90 dpv, 77% of the camels had developed antibodies. However, by 180 dpv only 37% of camels retained these antibodies. In addition, the average titer of virus-neutralizing antibodies (VNA) for each vaccinated group (see Appendix A) and the titer of their antibodies are expressed as log2.

### 3.4. Evaluation of the Reactogenicity and Take of the Live Attenuated Camelpox Vaccine

On the fifth day post vaccination, the overall health of the animals appeared to be satisfactory. Signs of vaccination were present in approximately 70–80% of the immunized camels and were visible as minor scarification at the administration site, but no other skin changes were noted (Figure 3, first and second rows). The body temperature of all animals fell within the normal range.

On the 12th day after immunization, vesicles (~95%) and poxvirus crusts (95–98%) formed at all vaccine scarification sites, as demonstrated previously with this vaccine [18]. This result indicated successful vaccine inoculation (Figure 3, 3rd and 4th rows).

### 3.5. Evaluation of Cellular Immunity Elicited following Vaccination Using a Delayed-Type Hypersensitivity Test (DTH)

DTH was carried out 28 days post vaccination using five randomly selected camels from each vaccinated group (Table 4). All the vaccinated camels, regardless of the vaccine type, exhibited a positive response to the hypersensitivity test, as indicated by their increased skin thickness. However, the thickness of the skin of control group of unvaccinated camels inoculated with non-infected cell culture homogenate did not increase at the injection site during testing. A two- to threefold increase in skin thickness occurred in vaccinated animals 24 h after inoculation with the inactivated CMLV antigen.

### 3.6. Vaccine Effectiveness

Vaccine effectiveness was determined based on the incidence rates observed in vaccinated versus unvaccinated camels. Two to three weeks post vaccination, symptoms of the disease manifested in the unvaccinated camels on these farms. One month later, camelpox started spreading more extensively. Fatalities among young camels and abortions among pregnant camels occurred. Based on the clinical signs and symptoms of the disease, local veterinary specialists made a preliminary diagnosis. Furthermore, this preliminary diagnosis was laboratory-confirmed by employees of the National Reference Center for Veterinary Medicine [12], and the virus was isolated from sick animals [20]. The vaccines successfully protected all the vaccinated camels from developing camelpox disease. Both the live and inactivated vaccines demonstrated 100% effectiveness (Table 5).

## 4. Discussion

The efficacy of a vaccine in field conditions is essential in demonstrating that the vaccine is effective. The field testing of both the live attenuated and inactivated camelpox vaccine candidates proved to be one of the timeliest anti-epidemic measures taken during the camelpox epidemic, helping prevent the spread of the camelpox epidemic to other regions. However, it was not possible to analyze the impact of our research results on camelpox epidemics in neighboring countries (Turkmenistan and Uzbekistan). The main reason for this is that there is no information about camelpox in the available literature from the aforementioned countries.

The Mangistau region borders Turkmenistan and Uzbekistan. Previously, when these countries were part of the USSR, large epizootics of camelpox occurred in Turkmenistan, which spread to the territory of the Kazakh SSR in 1965–1969, causing an epizootic in the Mangystau region [14]. In 2018 and 2019, in the Balkan velayat of Turkmenistan, bordering the Mangistau region of Kazakhstan, camelpox occurred, with deaths of both young animals and adults [26]. Although the cause of death of these camels was listed of famine, the clinical signs described strongly pointed to camelpox. Turkmenistan borders Iran, where outbreaks of this infection are recorded annually [27]. Sporadic cases and even epizootics of camelpox can occur in Uzbekistan, but no information is currently available. In Russia, the largest number of camels and their highest density are found in the Astrakhan region (up to 70% of all camels in the Russian Federation are concentrated here), which borders two regions of Kazakhstan: Atyrau and West Kazakhstan. In recent years, the situation regarding camelpox in the north of the Caspian Sea has been calm, but in the Soviet era, outbreaks of this infection were recorded here [28]. There are no official data on camelpox cases in China.

Safety testing for both vaccines was conducted as per the Guidelines for Diagnostic Tests and Vaccines for Terrestrial Animals (2022) [10] and Animal Safety Targets for Veterinary Live and Inactivated Vaccines [29]. Both vaccines were safe in camels, with camels vaccinated with the live attenuated vaccine displaying a local skin reaction, referred to as a take, with no adverse effects such as systemic reactions or changes in feed intake, behavior, or overall health. It is worth mentioning that the live vaccine did not affect the pregnancies of five camels in late gestation; their offspring were healthy at the end of the experiment. Overall, the vaccination of camels with both live and inactivated vaccines proved to be safe during field trials. There are limited studies on camelpox vaccines. In one study, both DucapoxR and inactivated camelpox vaccines were safe for the vaccination of pregnant camels [25]. Sudan CMLV/115 strain of CMLV was safe for young and pregnant camels [30]. At the same time, the Sudan CMLV/115 strain at a dose of 10^5.8^ TCID50/mL did not result in abortion or any other clinical signs in pregnant camels. In this study, the live attenuated vaccine tested on pregnant camels at a dose of 10^6.0^ did not lead to abortion or other clinical signs of the disease, and this dose was 1.3 times higher than that of the Sudan CMLV/115 strain.

Both the live and inactivated camelpox vaccines used the attenuated KM-40 strain in their manufacture. These vaccines have demonstrated their ability to induce immune responses in field studies. Notably, the live vaccine elicited an immune response starting on the seventh day, and by the 28th day 99% of vaccinated camels had seroconverted. Furthermore, 98% of the camels that received the live vaccine through the scarification method developed a take at the injection site. The appearance of an inflammatory skin process at the site of application is a protective reaction of the body, as veterinary legislation stipulates that, on the 5–12th day after vaccination, pustules should form at the site of scarification in animals. In the absence of a take, revaccination occurred within a specified timeframe [31]. The use of scarification for smallpox vaccinations in humans was able to eradicate the disease. This was also the case in small ruminants immunized against another parapoxvirus infections, such as contagious ecthyma in sheep and goats [32]. Moreover, the reversion to virulence of the KM-40 strain did not occur in camels after three serial passages, confirming its inability to revert to virulence. In addition, it was also deemed safe following an administered overdose. These results confirm that the developed vaccine does not lead to the development of clinical disease.

With the inactivated vaccine, serocoversion occurred in 77% of the camels at 90 days post vaccination. This rate of seroconversion was significantly lower than that of the live vaccine (*p* ≥ 0.05).

Both vaccines exhibited 100% efficacy during the camelpox outbreak, likely due to a combination of antibody and/or cellular immunity. Cellular immunity was demonstrated by a thickening of the skin in camels vaccinated with either vaccine, resulting in a delayed hypersensitivity reaction. These results align with those of similar studies [25]. In the days of the Soviet Union, the vaccinia virus also demonstrated 100% effectiveness against camelpox in field studies conducted in the region now known as Atyrau using skin scarification [14]. Notably, most vaccinated animals (approximately 95%) developed a benign, strictly localized rash at the vaccination site after 3–5 days, which resolved within 15–20 days. According to the same author [14], not a single case of camelpox infection occurred among the vaccinated camels during the observation period, while 85 cases occurred among non-vaccinated camels.

In this study, the immunogenicity of the inactivated vaccine was reproducible, as shown in a previous laboratory test [19]. Although the antibody response was very low, it showed 100% protection efficiency during a natural camelpox outbreak. In many studies, an experimental infection challenge demonstrates the effectiveness of vaccines. However, in our field tests, only a natural infection occurred, due to the lack of a specially equipped room in the region of the field test. The transportation of animals from an infected area cannot occur under the veterinary laws of the Republic of Kazakhstan [33]. However, the protective properties of these vaccines have been previously demonstrated in experimental studies [18,19]. Another limitation of this study was the lack of a proper sampling design to collect samples for camelpox virus shedding and serology following the detection of clinical signs in control camels. Since the purpose of camelpox virus vaccines are to prevent the severe clinical disease caused by camelpox, the fact that that no vaccinated camels experienced disease and a high percentage of non-vaccinated camels developed clinical disease demonstrates the vaccinations were effective.

The effect of vaccination on the camelpox virus outbreak is not known. It is unlikely that there was any effect on the early time points of the serological study, since the outbreak occurred 2–3 weeks after vaccination. However, it is possible that the infection of vaccinated camels could have increased their antibody responses following exposure. With capripoxviruses, an anamnestic antibody response does not occur in vaccinated animals compared to non-vaccinated animals due to a lack of virus replication driving the antibody response [34]. In addition, there were still differences observed between the antibody responses to the live attenuated and inactivated camelpox vaccines, indicating that these vaccines elicited different antibody levels despite these camels’ exposure to camelpox virus.

In field studies conducted by Khalafalla and El Dirdiri, the antibody responses in vaccinated camels were stable, ranging from 1:4–1:32 for up to one year [25]. In another study, all animals vaccinated with the Sudan/CMLV115 strain had a specific antibody response in the first week after vaccination, which ranged from 2 to 3 (log2) [30]. Similar results occurred using the Jouf-78 strain [35]. However, in both of the above field trials, the presence of virus-neutralizing antibodies occurred in animals before vaccination. In addition, in these field tests, the VNT used a small dose of virus equal to 10 TCID_50_, whereas, in our tests, the level of virus used in the VNT was 10 times higher, as per the guidelines for the methodology given by WOAH [10,24]. The low titer from the VNTs in our experiments may be due to the dose of the virus used in the VNTs. In addition, it is known that the inoculation of a live virus into the body leads to reproduction and, accordingly, to an increase in the antigen load, which affects the long-term formation of an immune response compared to the use of an inactivated virus, where the immune response depends on the amount of inactivated virus antigen introduced.

In this regard, the results of our research on the inactivated vaccine showed that this vaccine still has shortcomings, including the need for two vaccinations and a lower level and duration of antibody responses compared to the live attenuated vaccine. In addition, the technology for the preparation of the inactivated vaccine still needs to be improved and it is more expensive compared to the live attenuated vaccine. Following successful vaccine field trials in late 2020, the country initiated a program to immunize the camel population using a locally produced live attenuated vaccine [28]. To date, there has been over 300,000 vaccine doses produced for mass vaccination. This effort aims to create a barrier against camelpox, particularly in the Mangystau, Atyrau, and Aktobe regions.

## 5. Conclusions

Our field tests of candidate vaccines demonstrated that both vaccines were effective and contributed to the prevention of the spread of the camelpox epidemic to other regions. In addition, the appearance of pox rashes on the scarified skin during the use of the live vaccine indicates the immunogenicity of the vaccine, and this formation may mean that the use of additional methods to prove the protection of the vaccine are not required. The low-level antibody response elicited by the inactivated camelpox vaccine suggests that low levels of antibodies are sufficient to protect camels against camelpox. In addition, this study showed that locally produced live and inactivated vaccines were effective during the camelpox epidemic. Having these locally produced vaccines available is necessary for Kazakhstan to respond to new outbreaks.

## Figures and Tables

**Figure 1 vaccines-12-00685-f001:**
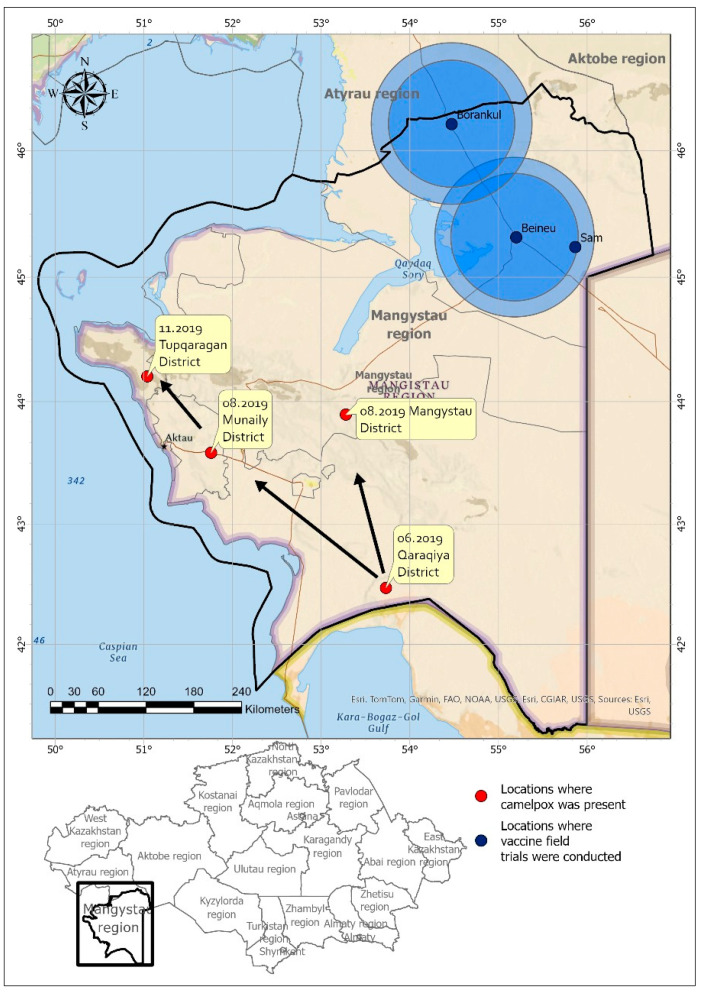
Map of the Mangystau region where the vaccine field test occurred. Red circles indicate the locations and dates of the outbreak of camelpox. The outbreak of camelpox first started in the Qaraqiya, Munaily, and Mangystau districts and then spread to other districts. Dark blue circles indicate areas where live and inactivated vaccines were tested in the field.

**Figure 2 vaccines-12-00685-f002:**
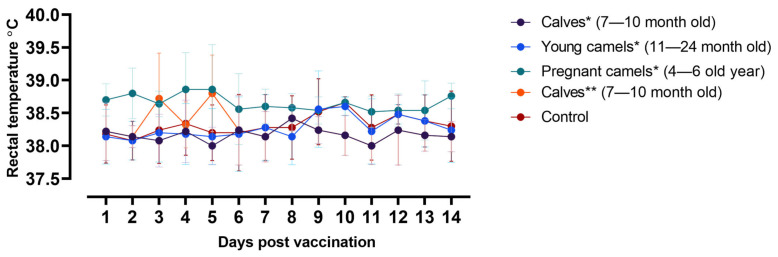
Temperature reaction of vaccinated animals after the administration of live N = 15 and inactivated vaccines N = 10 during the observation period. (*)—camels inoculated with live vaccine; (**)—camels inoculated with inactivated vaccine.

**Figure 3 vaccines-12-00685-f003:**
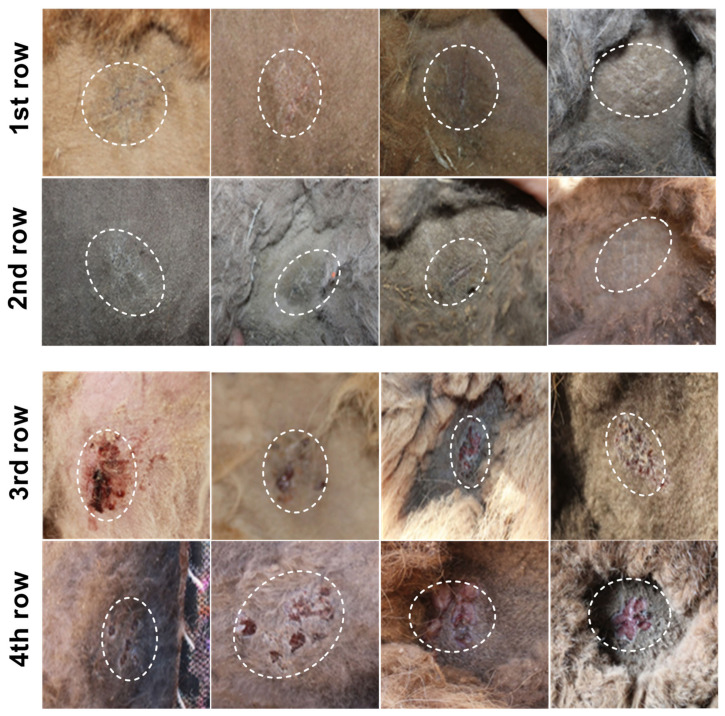
Immunogenicity of live camelpox vaccine in terms of the scarified skin of camels. The first and second rows show the results of the appearance of the scarified area of camel skin on the fifth day after vaccination. The third and fourth rows show the changes in the scarified skin of the camels on the 12th day after vaccination.

**Table 1 vaccines-12-00685-t001:** Vaccination protocol for live and inactivated camelpox vaccines.

Vaccine Type	Research Protocol	Vaccine Dose	Name of Locality and Number of Vaccinated Animals		
			Beineu	Borankul	Sam
Live	Immunogenicity	5 × 10^4.0^	27	52	38
	Safety	10^6.0^	5	5	5 **
Inactivated *	Immunogenicity	5 mL	-	-	30
	Safety	10 mL	-	-	10

Notes: (*)—The inactivated vaccine was tested only on young camels in the village of Sam. (**)—Pregnant camels in the last trimester of pregnancy.

**Table 2 vaccines-12-00685-t002:** Baseline serological survey of the field trial area.

Name of Rural District	Age and Number of Camels Studied			Seroprevalence Results
	7–9 Months of Age	10–13 Months of Age	≥14 Months of Age	
Beineu	27	15	38	negative
Borankul	18	8	45	negative
Sam	33	19	57	negative
TOTAL	78	42	140	

**Table 3 vaccines-12-00685-t003:** Seroconversion of the monitored camels after their vaccination with the live attenuated or inactivated camelpox vaccine in the Mangystau region across each day of follow-up.

Vaccine Type	VNA Titer via Serial Dilution	Days of Follow-Up					
		7	14	21	28	90	180
Live vaccine	1:2	3.42% (4/117)	11.1% (13/117)	45.3% (53/117)	11.1% (13/117)	-	23.1% (27/117)
	1:4	-	1.71% (2/117)	29.1% (34/117)	49.5% (58/117)	41.8% (49/117)	39.3% (46/117)
	1:8	-	-	17.9% (21/117)	38.4% (45/117)	38.4% (45/117)	28.2% (33/117)
	1:16	-	-	-	-	13.6% (16/117)	-
	1:32	-	-	-	-	2.56% (3/117)	-
	1:64	-	-	-	-	1.71% (2/117)	-
	1:128	-	-	-	-	0.85% (1/117)	-
**Total**		3.42% (4/117)	12.8 (15/117)	92.3 (108/117)	99 (116/117)	99 (116/117)	90.6 (106/117)
Inactivated vaccine	1:2	-	-	20.0% (6/30)	6.6% (2/30)	26.6% (8/30)	23.3% (7/30)
	1:4	-	-	6.6% (2/30)	23.3% (7/30)	30.0% (9/30)	10.0% (3/30)
	1:8	-	-	-	3.33% (1/30)	16.6% (5/30)	3.33% (1/30)
	1:16	-	-	-	-	3.33% (1/30)	-
	1:32	-	-	-	-	-	-
	1:64	-	-	-	-	-	-
	1:128	-	-	-	-	-	-
**TOTAL**		-	-	23.6 (8/30)	33.2 (10/30)	76.5 (23/30)	36.6 (11/30)

Notes: (-)—absence of immune response. The number of animals is indicated in parentheses, of which the numerator is the number of animals that showed an immune response to the CMLV, and the denominator is the total number of camels in the experiment.

**Table 4 vaccines-12-00685-t004:** Delayed-type hypersensitivity test (DTH) in camels treated with live and inactivated vaccines after the 28th dpv.

Location	Animal Status	Mean Skin Thickness (mm, M ± SD) Hours after Inoculation					
		0	24	48	72	96	120
Beineu	Vaccinated *	0.8 ± 0.07	1.3 ± 0.18	2.2 ± 0.44	2.4 ± 0.21	2.4 ± 0.17	2.0 ± 0.10
Boranqul	Vaccinated *	0.7 ± 0.15	1.2 ± 0.19	1.8 ± 0.37	2.2 ± 0.23	2.0 ± 0.30	1.7 ± 0.41
Sam	Vaccinated *	0.8 ± 0.11	1.1 ± 0.19	1.9 ± 0.39	2.1 ± 0.22	2.1 ± 0.08	1.9 ± 0.08
	Vaccinated **	0.8 ± 0.07	1.1 ± 0.16	1.4 ± 0.18	1.5 ± 0.16	1.5 ± 0.14	1.5 ± 0.15
	Unvaccinated (*n* = 5)	0.8 ± 0.08	1.0 ± 0.05	1.0 ± 0.05	1.0 ± 0.05	0.8 ± 0.05	0.8 ± 0.05
	*p* values ^#^	≥0.27–0.81	≤0.0001–0.03	≤0.0001	≤0.0001	≤0.0001	≤0.0001

Notes: (*)—group vaccinated with live vaccine; (**)—group vaccinated with inactivated vaccine; (#)—*p* value compared to non-vaccinated group.

**Table 5 vaccines-12-00685-t005:** The effectiveness of vaccines and comparative data on the incidence of camelpox in camels on farms in the Mangystau region between camels vaccinated with either a live attenuated or inactivated CMLP vaccine and non-vaccinated camels (control).

Location	Animal Status	Camel Count	Sick		Dead		Aborted	Vaccine Efficacy, %
			Young	Adults	Young	Adults		
Beineu	Vaccinated *	27	0	0	0	0	0	100
	Unvaccinated	123	46	67	8	7	4	-
Boranqul	Vaccinated *	52	0	0	0	0	0	100
	Unvaccinated	120	17	53	0	3	1	-
Sam	Vaccinated *	38	0	0	0	0	0	100
	Vaccinated **	30	0	-	0	-	-	100
	Unvaccinated	115	11	47	5	2	3	-
TOTAL		505	74	167	13	8	8	

Notes: (*)—group vaccinated with live vaccine; (**)—group vaccinated with inactivated vaccine.

## Data Availability

Data is contained within the article or Appendix A.

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
