# Peer review of "Field Trials of Live and Inactivated Camelpox Vaccines in Kazakhstan"

_vaccines, 2024, doi:10.3390/vaccines12060685_

Round 1
Reviewer 1 Report
Comments and Suggestions for Authors
Overall Comments:
This study explores field trials of live and inactivated camelpox vaccines carried out by the Research Institute for Biological Safety Problems in Kazakhstan. The efficacy of these vaccines was assessed through trials on 172 camels in the Beineu district. 132 camels received the live attenuated vaccine, while 40 received the inactivated version to assess immunogenicity and safety. Both vaccines showed no adverse effects on the camels post-vaccination. The live vaccine prompted antibody responses in some animals by day seven, with 99% responding by day 28. In contrast, camels immunized with the inactivated vaccine showed seroconversion starting on day 21 at lower titers, with 77% showing an immune response 90 days post-vaccination. The antibody response decreased after six months. Both vaccines proved 100% effective in preventing clinical disease during a camelpox outbreak; unvaccinated camels fell ill. Subsequent to successful trials in Kazakhstan, a vaccination initiative employing domestically produced live attenuated camelpox vaccine has commenced to control camelpox in camels.
To strengthen the manuscript, I suggest the following revisions:
1) The background introduction in the abstract is too brief and lacks detail. You need to provide a more comprehensive overview of the topic to engage readers.
2) Provide more context and background information on the topic to make the article more accessible to a wider audience.
3) Address the limitations of the study, such as the reliance on biased datasets and the absence of a proper sampling design. Discuss potential implications of these limitations on the interpretation of the findings.
4) The article could benefit from a thorough proofreading and editing to improve the clarity and coherence of the writing. This includes addressing any grammatical errors, improving sentence structure, and ensuring consistency in terminology and formatting.
Comments on the Quality of English LanguageModerate editing of English language required.
Author Response
Dear Reviewer,
All your comments have been answered. You can find the answers in the attached file.

Reviewer 2 Report
Comments and Suggestions for Authors
In this study, Muratbay Mambetaliyev and colleagues analyzed the field trials of live and inactivated camelpox vaccines conducted in Kazakhstan. Both vaccines were determined to be 100% effective in preventing clinical disease in vaccinated camels during the outbreak of camelpox. This study holds significant importance for the control of camelpox outbreaks in Kazakhstan. Despite its clear structure and comprehensive analysis, this manuscript lacks clarity on some important issues.
Major issues:
1. Please clarity the dose of administration for immunization in the abstract.
2. How to quantify attenuated live vaccines and inactivated vaccines needs to be explained in detail.
3. The phrase "without any problems" in line 426 is too absolute, it is suggested to modify the wording.
4. Please rewrite the conclusion section to highlight the results of this study.
5. Has the effective storage time of the vaccine been evaluated?
6. Please provide detailed background information about KM-40 strain, such as its genome accession number and median lethal dose.
Author Response

(The authors gave the same response as above.)

Reviewer 3 Report
Comments and Suggestions for Authors
My views on the article titled "Field trials of live and inactivated camelpox vaccines in Kazakhstan" are given below:
1. Line 115: Please add the concentration of β-propiolactone used for inactivation.
2. Line 154: Before the second vaccination, was a small amount of blood collected to confirm the antibody levels in the camels after the primary immunization?
3. Line 168: 106EID50/ml should be 106EID50/ml.
4. Table 2: Is "≤ 14 months of age" an editorial error, or does it specifically refer to a certain age group? How does it differentiate from the previous "7-9 months of age" and "10-13 months of age"?
The research is highly significant for the prevention of camelpox infection. By randomly selecting camels of different age groups for vaccination with either the live attenuated vaccine or the inactivated vaccine, and comparing them with unvaccinated camels, the study evaluates the effectiveness and safety of both vaccines in preventing camelpox infection. Although the antibody levels produced by the inactivated vaccine, even with the addition of adjuvant, were still lower and appeared later than those produced by the attenuated live vaccine, both vaccines provided 100% protection for the camels. This research provides strong data support for the prevention of camelpox infection in camels.
I congratulate the authors on their successful work.
Author Response

(The authors gave the same response as above.)

Round 2
Reviewer 1 Report
Comments and Suggestions for Authors
The authors have addressed my comments.